# Receptor-mediated dimerization of JAK2 FERM domains is required for JAK2 activation

Ryan D Ferrao, Heidi JA Wallweber, Patrick J Lupardus*

Department of Structural Biology, Genentech, Inc., South San Francisco, United States

**Abstract** Cytokines and interferons initiate intracellular signaling via receptor dimerization and activation of Janus kinases (JAKs). How JAKs structurally respond to changes in receptor conformation induced by ligand binding is not known. Here, we present two crystal structures of the human JAK2 FERM and SH2 domains bound to Leptin receptor (LEPR) and Erythropoietin receptor (EPOR), which identify a novel dimeric conformation for JAK2. This 2:2 JAK2/receptor dimer, observed in both structures, identifies a previously uncharacterized receptor interaction essential to dimer formation that is mediated by a membrane-proximal peptide motif called the 'switch' region. Mutation of the receptor switch region disrupts STAT phosphorylation but does not affect JAK2 binding, indicating that receptor-mediated formation of the JAK2 FERM dimer is required for kinase activation. These data uncover the structural and molecular basis for how a cytokine-bound active receptor dimer brings together two JAK2 molecules to stimulate JAK2 kinase activity.

DOI: https://doi.org/10.7554/eLife.38089.001

## Introduction

Janus kinases (JAKs) are a family of multi-domain non-receptor tyrosine kinases responsible for pleiotropic regulatory effects on growth, development, immune and hematopoietic signaling (*Leonard and O'Shea, 1998*). The JAK family consists of four conserved members, including JAK1, JAK2, JAK3, and TYK2, which are differentially activated in response to cytokine and interferon stimulation. JAKs are constitutively bound to the intracellular domains of their cognate cytokine signaling receptors, and are activated after cytokine-mediated dimerization or rearrangement of these receptors establishes a productive receptor signaling complex (*Haan et al., 2006*). The canonical JAK signaling pathway initiates with kinase trans-autophosphorylation, followed by phosphorylation of receptor intracellular domains, recruitment and phosphorylation of STAT transcription factors, and translocation of active STAT dimers to the nucleus to initiate transcription of target genes. In the quarter century since their discovery, the JAKs and their cognate cytokines and receptors have emerged as critical drug targets for immune disorders as well as cancer (*Kontzias et al., 2012*; *Liu et al., 2013*).

JAKs share a conserved four domain structure, with each domain playing a distinct and understood role in JAK function. The C-terminal half of the archetypical JAK contains hallmark tandem pseudokinase and kinase domains, with the C-terminal tyrosine kinase domain essential for its enzymatic activity, and the pseudokinase playing a role in regulation of the kinase domain (*Saharinen et al., 2000*; *Saharinen and Silvennoinen, 2002*). At the N-terminus, sequential FERM and SH2 domains are responsible for distinct receptor interactions. The FERM domain is itself made up of three subdomains, including a ubiquitin-like fold (F1), an acyl CoA-binding protein-like domain (F2), and a Plextrin Homology (PH)-like fold (F3). These three domains form an interwoven

*For correspondence:
lupardus.patrick@gene.com

cloverleaf-like structure and are closely associated with the SH2 domain to form a receptor binding holodomain (*Ferrao and Lupardus, 2017*). Prior structures of the FERM–SH2 module have identified several receptor peptide-binding sites within the FERM–SH2, including a 'box1' binding site on the FERM F2 subdomain, and a second 'box2'-binding site on the SH2 domain (*Wallweber et al., 2014*; *Ferrao et al., 2016*; *Zhang et al., 2016*).

JAK2 is in many ways the prototypical member of the JAK family, with an essential signaling role for cytokines and interferons involved growth and energy homeostasis (HGH, Leptin), hematopoiesis (GMCSF, EPO, TPO, IL-3), immunity and allergy (IL-12, IL-23, IL-5), and antiviral responses (IFNγ) (*Babon et al., 2014*). JAK2 ablation in mice results in embryonic lethality due to disruption of eryth-ropoiesis *in utero* (*Neubauer et al., 1998*), underlying a critical need for JAK2 signaling in the development of the hematopoietic system. JAK2 is also a proto-oncogene, with constitutively activating pseudokinase mutations such as V617F shown to drive a subset of myelo- and lympho-proliferative disorders (*Vainchenker and Constantinescu, 2013*).

As the diversity of JAK2-dependent cytokines suggests, JAK2 activity can result from ligation of a number of homodimeric and heterodimeric pairs of signaling receptors. While JAK2-activating heter-odimeric pairs are found in both the class I and class II cytokine receptor families (*Ihle et al., 1995*; *Wang et al., 2009*), receptors that utilize JAK2 in a homodimeric assembly are a smaller group and fall into two class I subfamilies: the growth hormone family (*Ihle et al., 1995*) and the 'tall' receptor family (*Wang et al., 2009*). Erythropoietin receptor (EPOR) is emblematic of the first group, and includes a canonical extracellular cytokine binding homology region (CHR) motif consisting of tandem FNIII-like domains that form a 2:1 complex with a single EPOR molecule (*Syed et al., 1998*). In the tall receptor family, only gp130 and the leptin receptor (LEPR) have the ability to homodimerize in response to cytokine binding (*Waters and Brooks, 2015*). Gp130 and LEPR contain six and seven Ig/FNIII-like domains, respectively, and require two cytokines (i.e. IL-6 or LEPR) for assembly of the signaling homodimer (*Boulanger and Garcia, 2004*; *Mancour et al., 2012*).

While cytokine-induced dimerization of dispersed monomeric receptors is the canonical model of cytokine signaling, mounting evidence suggests that at least a subset of receptors can exist in a pre-dimerized state, and that conformational shifts in the transmembrane (TM) helices are a switch that initiates JAK activation (*Seubert et al., 2003*; *Brooks et al., 2014*; *Matthews et al., 2011*; *Defour et al., 2013*). If cytokine-induced conformational change in the receptor dimer is indeed the trigger for JAK activation, we hypothesized that these changes may produce a dimeric JAK conformation that brings together the kinase domains to initiate trans-phosphorylation and downstream signaling. We therefore set out to structurally characterize the JAK2 FERM–SH2 bound to peptides from homodimeric signaling receptors, with the goal of capturing the membrane-proximal domains of JAK2 in an activated dimeric state.

## Results

### Crystal structures of the JAK2 FERM-SH2 bound to EPOR and LEPR

JAK FERM–SH2 domains have been successfully crystallized with their receptors by either fusing the JAK-binding receptor fragment to the C-terminus of the SH2 domain (*Wallweber et al., 2014*; *Zhang et al., 2016*), or by co-expression of GST-receptor fusion proteins with the JAK FERM–SH2 (*Ferrao et al., 2016*). To obtain structures of JAK2 with a homodimeric receptor, we utilized both methods across a number of receptors to identify the best possible samples for crystallography, with the goal of having a representative structure from both class I receptor subfamilies, the growth hormone family and the tall receptor family. Ultimately, we identified two receptors, EPOR and LEPR (*Figure 1A*), that produced well-behaved complexes with JAK2 for crystallization trials. For the JAK2/EPOR complex, fusion of the cytoplasmic box1 and box2 containing fragment of EPOR (Ser273-Cys338) to the C-terminus of the human JAK2 FERM–SH2 (Asp36-Thr514) resulted in a protein complex that was stable and purified to high yield. During the final stages of purification, this JAK2/EPOR fusion spontaneously crystallized at neutral pH. To decrease the rate of spontaneous nucleation and allow for further concentration for crystallization trials, the pH was lowered to pH 5.5 and the protein subjected to crystallization screening. For crystallization of JAK2 with LEPR, we generated a LEPR construct containing the predicted Box1 and Box2 domains (Ser863-Glu933) with an N-terminal GST fusion tag and a TEV protease site for co-expression with the JAK2 FERM–SH2.

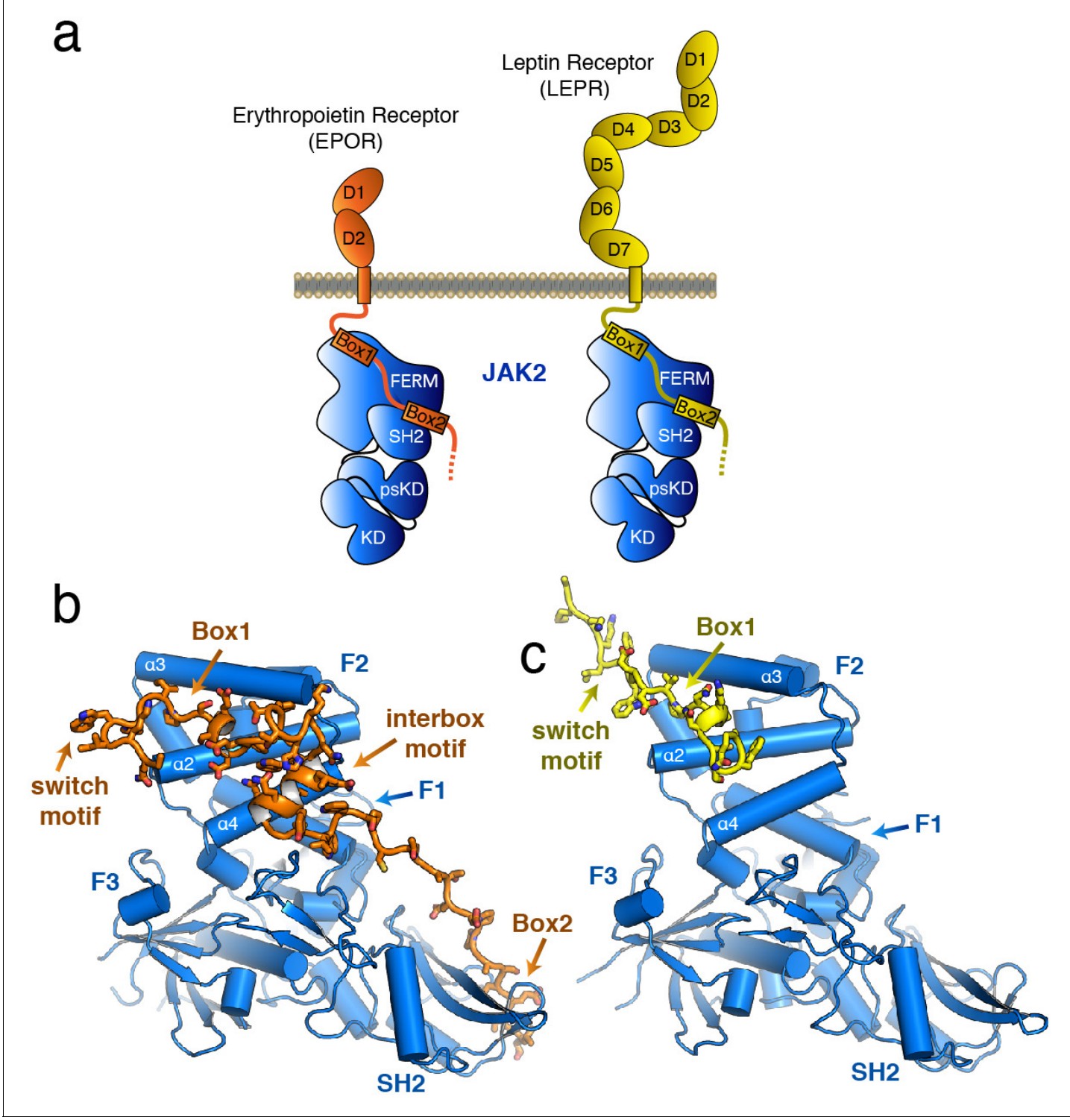

**Figure 1.** The structure of the JAK2 FERM–SH2 domain bound to EPOR and LEPR. (a) Schematic representation of erythropoietin receptor (EPOR) and leptin receptor (LEPR) bound to JAK2. Each receptor binds to JAK2 via a box1 interaction with the FERM domain, and a box2 interaction with the SH2 domain. (b,c) The crystal structures of EPOR and LEPR bound to JAK2 at 2.65 and 2.83 Å respectively. (b) Cartoon representation of residues 279 to 335 of EPOR bound to JAK2 FERM–SH2. JAK2 is shown in blue, and EPOR shown in orange. (c) Cartoon representation of residues 866 to 885 of LEPR bound to JAK2. JAK2 is shown in blue, and LEPR shown in yellow.

DOI: https://doi.org/10.7554/eLife.38089.002

The following figure supplement is available for figure 1:

*Figure 1 continued on next page*

*Figure 1 continued*

**Figure supplement 1.** Sequence and electron density for JAK2 bound receptors EPOR and LEPR.
DOI: https://doi.org/10.7554/eLife.38089.003

Following TEV cleavage to remove GST, the JAK2/LEPR complex was purified, concentrated, and subjected to crystallization screening. Ultimately, we were able to obtain complete native datasets for JAK2/EPOR and JAK2/LEPR that diffracted to 2.65 and 2.83 Å, respectively (*Table 1*).

The overall structure of the JAK2 FERM–SH2 module in both structures is similar to the 'apo' JAK2 FERM–SH2 structure (*McNally et al., 2016*), with a root mean square deviation (RMSD) for both structures of approximately 0.5 Å over 333 Cα atoms (*Figure 1B–C*). The FERM domain adopts the expected tri-lobed architecture, comprised of three subdomains known as F1, F2, and F3. The SH2 domain of JAK2 packs against the F1 and F3 subdomains and is held in place by an elongated linker between the F3 and SH2 domains as well as a linker C-terminal to the SH2 domain. For the EPOR complex with JAK2, unambiguous electron density was present for both the box1 and box2 motifs of the EPOR receptor, along with a novel 22 residue segment in between box1 and box2 that we call the 'interbox' motif (*Figure 1B*). In all, the model contains EPOR residues 279 to 335. For the LEPR complex with JAK2, electron density was visible for residues 866 to 885, which includes the

**Table 1.** Data collection and refinement statistics.

|  | JAK2/EPOR | JAK2/LEPR |
|---|---|---|
| Data collection | ALS 5.0.1 | SSRL 12–2 |
| Space group | C2 | P6$_5$22 |
| Cell dimensions |  |  |
| $a$, $b$, $c$ (Å) | 178.49, 114.88, 179.82 | 263.87, 263.87, 101.08 |
| α, β, γ (°) | 90, 93.2, 90 | 90, 90, 120 |
| Resolution (Å) | 48.44–2.65 (2.74–2.65) | 43.19–2.83 (2.93–2.83) |
| $R_{sym}$ or $R_{merge}$ | 0.073 (0.865) | 0.105 (1.60) |
| $I$ / σ$I$ | 13.1 (1.3) | 21.8 (2.0) |
| Completeness (%) | 99.5 (97.9) | 99.6 (99.5) |
| Redundancy | 3.4 (3.3) | 13.4 (13.7) |
| CC1/2 | 0.99 (0.63) | 0.99 (0.80) |
| Refinement |  |  |
| Resolution (Å) | 48.44–2.65 (2.75–2.65) | 43.19–2.83 (2.93–2.83) |
| No. reflections | 104,921 (10,233) | 49,498 (4853) |
| $R_{work}$/$R_{free}$ | 0.222/0.260 | 0.228/0.241 |
| No. atoms | 16599 | 7601 |
| Protein | 16454 | 7569 |
| Ligand/ion | N/A | 5 |
| Water | 145 | 27 |
| $B$-factors | 71.32 | 106.19 |
| Protein | 71.48 | 106.30 |
| Ligand/ion | N/A | 104.80 |
| Water | 53.65 | 74.23 |
| R.M.S. deviations |  |  |
| Bond lengths (Å) | 0.003 | 0.004 |
| Bond angles (°) | 0.85 | 0.63 |

Values in parentheses are for highest-resolution shell.
DOI: https://doi.org/10.7554/eLife.38089.004

box1 motif as well as several residues N-terminal to the box1 (*Figure 1C*). Weak electron density for a LEPR interbox motif as well as the box2 was visible, but the quality of the density was too poor to build an acceptable model (*Figure 1—figure supplement 1*).

## EPOR and LEPR interactions with the JAK2 FERM

The box1 motif of class I cytokine receptors is defined by a conserved φ-Pro-X-Pro motif shared by nearly all family members (*Figure 2A* and *Figure 1—figure supplement 1*). The EPOR and LEPR box1 motifs begin with similar aliphatic residues (EPOR Ile286 and LEPR Val875), followed by the shared Pro-X-Pro motif that places both proline residues into a groove formed by the F2-α2 and F2-α3 of JAK2 (*Figure 2B–C*). Downstream of the φ-Pro-X-Pro motif, there is a short pseudo-helical turn in both EPOR and LEPR, positioning either EPOR Ser291 or LEPR Asn880 to hydrogen bond with JAK2 Glu176. Additional interactions are picked up between EPOR Phe293 or LEPR Trp883, which share a hydrophobic interaction surface on JAK2 near Leu184 and Phe240.

Comparison of these receptor-bound JAK2 structures to the apo JAK2 FERM–SH2 structure (*McNally et al., 2016*) identifies a number of side chain rotamer movements within the F2 subdomain that accompany receptor binding (*Figure 2—figure supplement 1*). Further comparison of the EPOR and LEPR class I receptor peptides bound to JAK2 with the structure of IFNLR class II receptor peptide bound to JAK1 (*Ferrao et al., 2016*; *Zhang et al., 2016*) shows that the receptor-binding interface is similar between JAK1 and JAK2, with some notable differences (*Figure 2—figure supplement 1*). Interestingly, the interaction site on JAK2 for the first box1 proline residue in EPOR and LEPR is occupied by IFNLR Trp257 in the JAK1 structure, while a rotamer shift in JAK1 Phe247 (equivalent to JAK2 Phe236) that facilitates interaction with IFNLR1 Pro264 would prohibit binding of the φ-Pro-X-Pro motif (*Figure 2—figure supplement 1*). Given JAK1 is capable of binding some φ-Pro-X-Pro containing Class I receptors, the plasticity observed in the JAK1 and JAK2 F2 subdomain suggests significant rotamer adjustments may accompany binding to different classes of cytokine receptors.

In our JAK2/EPOR structure, we also find that the EPOR interbox region contributes a previously undescribed 22-residue folded mini-domain C-terminal to the JAK2/EPOR interaction (*Figure 2D*). The core of the interbox domain is a 2.5 turn alpha helix that packs several hydrophobic sidechains (Phe104, Trp307, and Leu308) against the JAK2 F2-α4 helix. At the center of the interbox domain lies Trp307, which interacts with nearby residues within the helix as well as Phe293 and Leu296 at the C-terminus of the box1 motif. At the terminus of the helix, Asp312 forms a salt bridge with JAK2 Lys253, followed by a tandem tryptophan motif (Trp316/317) that occupies two different conformations within the asymmetric unit of the crystal. After this tandem tryptophan motif, we see a 10 residue stretch of amino acids that make only tangential contact with JAK2. Contact is regained with JAK2 at Pro328, followed by the binding of Leu331 and Val333 into the canonical box2 binding groove on the SH2 domain (*Figure 2E*), originally described for the TYK2/IFNAR1 interaction (*Wallweber et al., 2014*).

## The EPOR and LEPR 'switch' motif residues bridge a JAK2 dimer

Although the JAK2/EPOR and JAK2/LEPR complexes crystallized in two distinct crystal forms (*Table 1*), similarities were immediately evident after analysis of packing interactions between JAK2 monomers in the crystal lattices. The JAK2/EPOR asymmetric unit contained four JAK2 and four EPOR molecules. Each asymmetric unit contained two nearly identical dimeric 2:2 JAK2/EPOR complexes (*Figure 1—figure supplement 1*) oriented around a pseudo-symmetric two-fold axis. The two JAK2 molecules interact through reciprocal contacts between the FERM F2 and F3 subdomains, with each receptor bridging a constitutively bound F2 subdomain to an opposing F3 subdomain (*Figure 3A–B*). For JAK2/LEPR, each asymmetric unit contained two JAK2 and two LEPR molecules, with each independent JAK2/LEPR monomer involved in an interaction with a crystallographic symmetry mate in a neighboring unit cell (*Figure 1—figure supplement 1*). These intermolecular packing contacts generate two nearly identical and symmetric 2:2 receptor/JAK dimer complexes that are topologically similar to the JAK2/EPOR dimer (*Figure 3C–D*).

The EPOR and LEPR fragments co-crystallized with JAK2 were designed to begin immediately after the transmembrane domain, and in both cases, residues within this membrane proximal receptor segment bridge the interaction with the second JAK2 molecule in the dimer. For EPOR, this

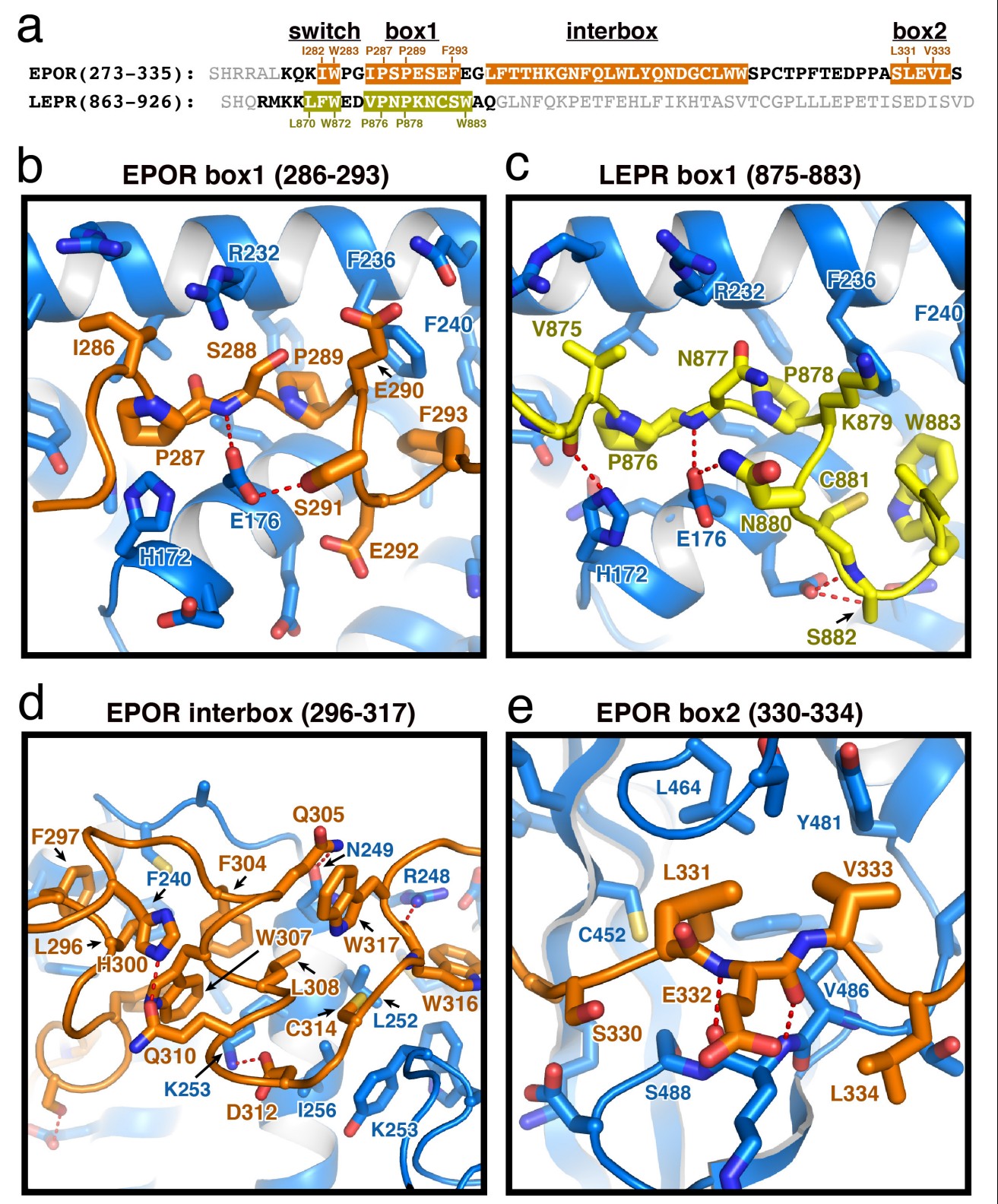

**Figure 2.** EPOR and LEPR interactions with JAK2. (a) Alignment of the intracellular receptor sequences of human EPOR and LEPR that interact with JAK2. Sequences begin at the first residue after termination of the transmembrane domain. Sequences were aligned using the φ-Pro-X-Pro motif as an anchor sequence. (b–e) Detailed views of interactions between JAK2 and (b) EPOR box1, (c) LEPR box1, (d) EPOR interbox region, and (e) EPOR box2.

*Figure 2 continued on next page*

*Figure 2 continued*

EPOR and LEPR are colored in orange and yellow, respectively, with amino acid side chains shown as sticks. JAK2 is colored blue, with amino acid side chains shown as sticks. Key residues are labeled for reference.

DOI: https://doi.org/10.7554/eLife.38089.005

The following figure supplement is available for figure 2:

**Figure supplement 1.** Structural analysis of receptor box1 interactions.

DOI: https://doi.org/10.7554/eLife.38089.006

segment has been previously branded the 'switch' region, with several residues within the segment required for JAK activation (*Constantinescu et al., 2001*; *Huang et al., 2001*). Both the JAK2/EPOR and JAK2/LEPR crystal structures show that this switch region interaction is mediated by hydrophobic residues inserting into the same pocket on the PH-like F3 subdomain of the JAK2 FERM. In addition, the switch residues in EPOR and LEPR are similarly positioned in sequence just N-terminal to the box1 motif (*Figure 2A*), an area that is highly enriched in aromatic residues in other cytokine receptors (*Figure 1—figure supplement 1*). For EPOR, the key switch region contact residues are Ile282 and Trp283 (*Figure 3B*), and for LEPR the key contact residues are Leu870, Phe871, and Trp872 (*Figure 3D*). The receptor-binding pocket on the F3 subdomain is created by the intersection of the β1-β4 sheet and β7 strand, and is lined on one side by a β3 strand tryptophan residue (JAK2 Trp298) that is conserved among JAK family members (*Figure 3—figure supplement 1*). EPOR Ile282 and Trp283 insert directly into this F3 pocket, while for LEPR Leu870 inserts into the pocket and Trp872 makes an edge-face π-π interaction with the opposite face of JAK2 Trp298. This pocket on the PH-like F3 subdomain is also the conserved interaction site for inositol phosphate headgroups in a number of classical PH domains (*Lemmon, 2007*), and facilitates dimerization of Focal Adhesion Kinase (FAK) FERM domains via a topologically similar tryptophan-mediated interaction (*Brami-Cherrier et al., 2014*) (*Figure 3—figure supplement 1*).

While the contact sites for the EPOR and LEPR switch residues on the JAK2 F3 subdomain are similar, the contacts between the JAK2 F2 and F3 subdomains are unique between the two complexes. This is primarily due to the different contact angles for each complex, with each JAK2 in the EPOR complex opposed at approximately 120°, versus each JAK2 in the LEPR complex opposed at approximately 180° (*Figure 3B,D*). In the JAK2/EPOR complex the one significant contact between JAK2 molecules is a salt bridge between Glu173 in F2 α2 helix, and Arg300 in the opposing F3 β3 strand (*Figure 3—figure supplement 1*). In the JAK2/LEPR structure, there are a number of contacts between the F2 and F3 subdomains (*Figure 3—figure supplement 1*). Instead of forming a salt bridge with the opposing JAK2 F2, Arg300 is folded back and forms an intra-domain ionic pairing with F3 Glu274. JAK2 F3 Asp313 forms a polar contact with the backbone amide of F2 His172, and the sidechains of His172 and His222 in the F2 subdomain intercalate into the interface between LEPR and the two JAK2 molecules (*Figure 3—figure supplement 1*). These differing JAK2 F2/F3 dimer contacts in the EPOR and LEPR structures highlight the permissiveness of the JAK2–JAK2 interface, and suggest the driving force for dimerization is receptor-mediated contact with the opposing JAK2 molecule. Importantly, this dimeric JAK2/receptor conformation was only observed in the context of a crystal lattice formed at high-protein concentration, and we were unable to reconstitute this dimer in solution. This is likely due to a low-affinity between the soluble JAK FERM–SH2/receptor monomers, which would normally encounter one another in the context of a membrane tethered and conformationally restricted receptor dimer.

## The switch motif is dispensable for JAK2 binding, but essential for STAT phosphorylation

Based on prior studies suggesting that the switch region is essential for signaling but not JAK binding (*Constantinescu et al., 2001*; *Huang et al., 2001*; *Haan et al., 2002*), we hypothesized that the JAK dimers observed in our crystal structures represent functional signaling complexes, with switch-mediated dimer formation required for JAK activation. To first interrogate the role of LEPR switch residues on binding to JAK2, we utilized BioLayer Interferometry (BLI) to alanine-scan the interaction between JAK2 and biotinylated LEPR peptides generated by in vitro translation. Wild-type LEPR peptide bound to JAK2 with an equilibrium affinity constant ($K_D$) of 18.7 ± 1.3 μM (*Figure 4* and

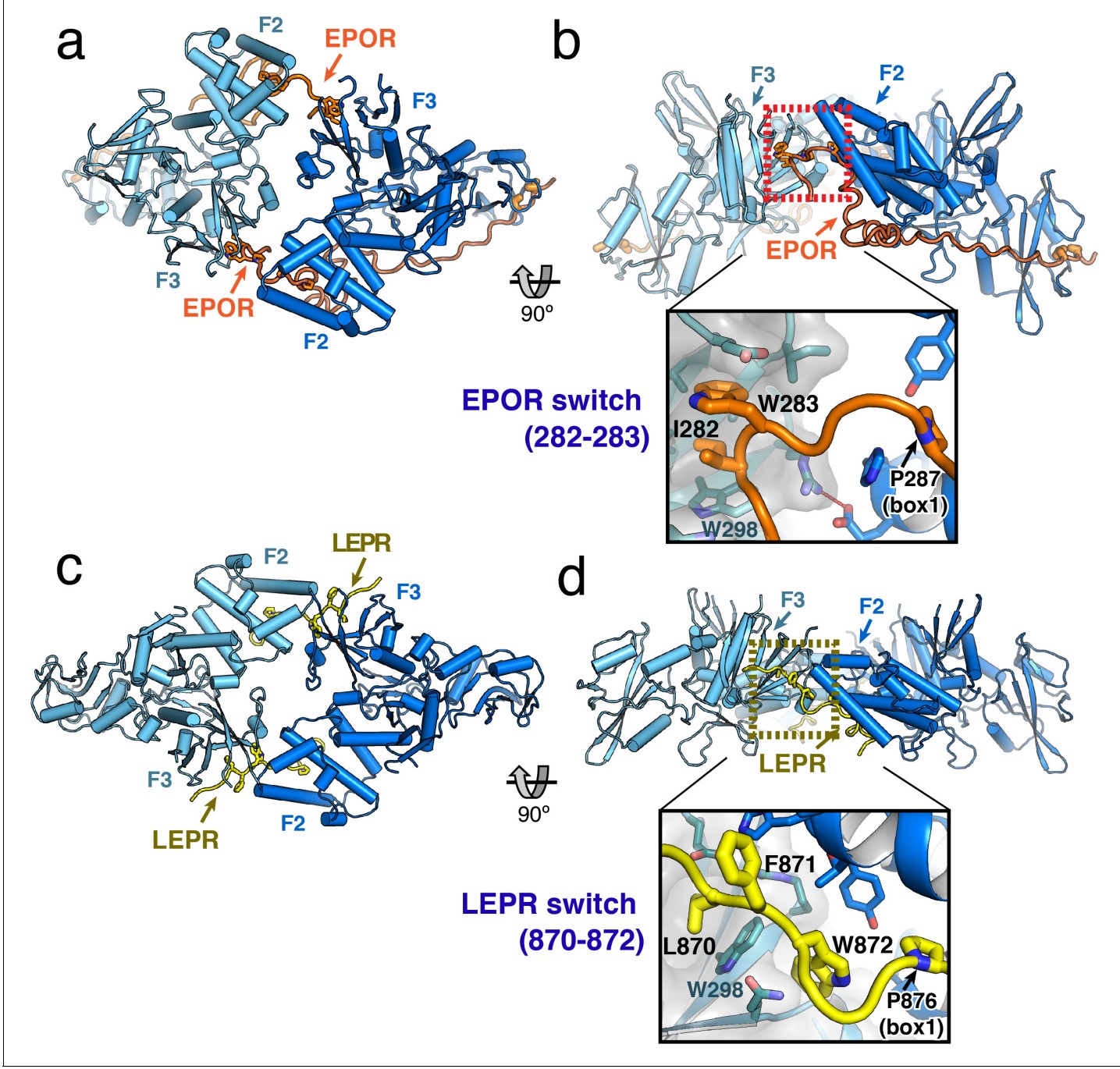

**Figure 3.** JAK2/EPOR and JAK2/LEPR dimerization is mediated by the receptor 'switch' regions. (a) Top and (b) side views of the JAK2/EPOR dimer displayed as a cartoon model, with JAK2 monomers shown in blue and teal, and EPOR shown in orange. Inset box in (b) shows a close-up view of the EPOR switch residues Ile282 and Trp283, shown as stick models. Box1 residue Pro287 is also shown for reference. (c) Top and (d) side views of the JAK2/LEPR dimer displayed as cartoon models, with JAK2 monomers shown in blue and teal as in (a) and (b), and with LEPR shown in yellow. As in (b), the inset box shows a close-in view of the LEPR switch residues 870–872, displayed as stick models. Pro876 from the LEPR box1 sequence is also shown for reference.

DOI: https://doi.org/10.7554/eLife.38089.007

The following figure supplement is available for figure 3:

**Figure supplement 1.** Structural analysis of receptor 'switch' region interactions with JAK2 F3 subdomain.

DOI: https://doi.org/10.7554/eLife.38089.008

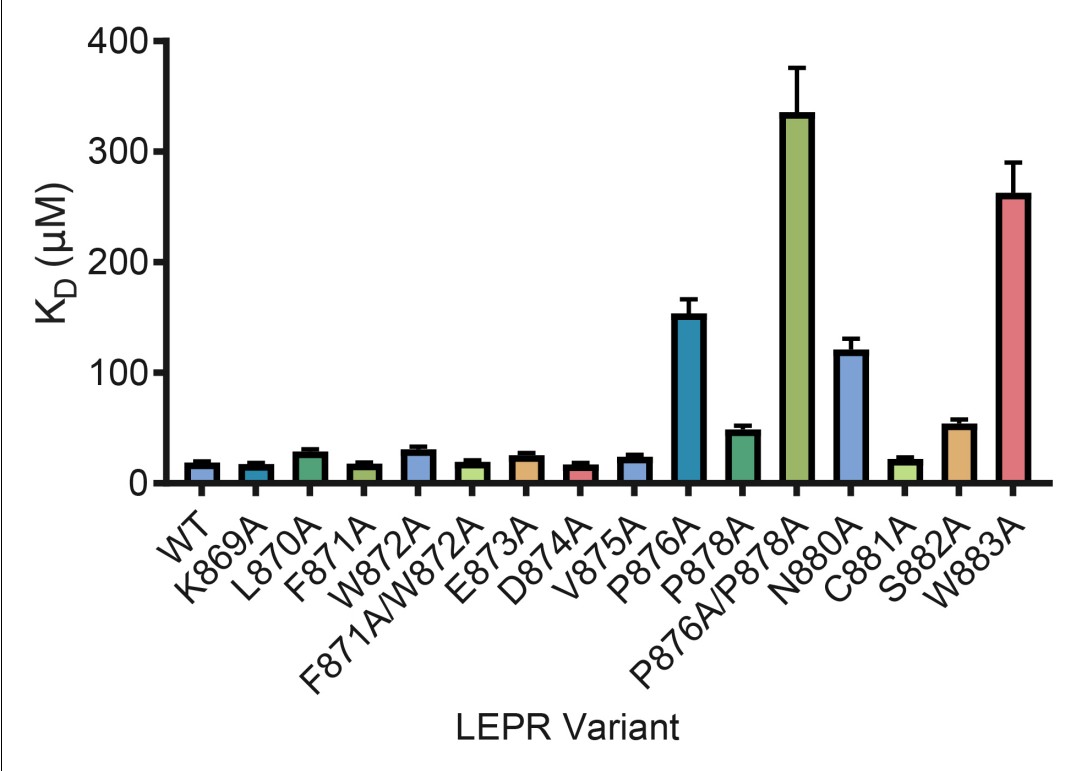

**Figure 4.** LEPR switch residues are dispensable for binding to JAK2. Biolayer Interferometry was used to measure equilibrium affinity constants ($K_D$) for the binding of in vitro translated human LEPR (residues 863–933) containing the listed mutations to wild-type JAK2 FERM–SH2 protein (residues 36–514). The $K_D$ ± standard error of three replicate experiments is represented as a bar graph.

DOI: https://doi.org/10.7554/eLife.38089.009

The following figure supplement is available for figure 4:

**Figure supplement 1.** Biolayer interferometry (BLI) analysis of the JAK2/LEPR interaction.

DOI: https://doi.org/10.7554/eLife.38089.010

*Figure 4—figure supplement 1*). When alanine point mutations in box1 residues Pro876, Pro878, Asn880, and Trp883 were tested, we found these mutations strongly reduced the affinity of JAK2 for LEPR. Yet when we tested alanine mutations in juxtamembrane residues between Lys869 and Asp874, including the switch contact residues Phe871 and Trp872, these peptides retained affinity for JAK2. These results indicate that the switch residues in LEPR are not required for JAK2 binding. We also attempted to perform these experiments on EPOR, but were unable to obtain appropriate levels of in vitro translation of EPOR or sufficiently solubilize synthesized EPOR peptides in aqueous buffer for BLI assays (*Figure 4—figure supplement 1*).

We then tested the effect of switch region and box1 alanine mutations on JAK2 activity in the murine Ba/F3 cell line. Stably transfected Ba/F3 cells expressing the murine leptin receptor (LEPR) or erythropoietin receptor (EPOR) were generated with mutations in the switch region or box1 motif. Importantly, human and murine EPOR and LEPR share 100% identity in the switch and box1 regions, with slightly altered residue numbering (*Figure 5—figure supplement 1*). Each cell line was assayed for surface expression of EPOR and LEPR by flow cytometry, and expression of both wild-type and mutant receptors was similar for all cell lines (*Figure 5—figure supplement 1*). Cells were then starved, stimulated with EPO or LEPR, and assayed for STAT5 or STAT3 phosphorylation by phospho-flow cytometry. Cells expressing wild-type EPOR exhibited increased levels of phosphorylated STAT5 (pSTAT5) when stimulated with EPO (*Figure 5A*), and likewise, stimulation of cells expressing wild-type LEPR with leptin resulted in increased levels of phosphorylated STAT3 (pSTAT3) (*Figure 5b*).

As expected, when cells expressing box1 mutations in EPOR (P286A/P288A) or LEPR (P874A, P874A/P876A, W881A) were stimulated with EPO or Leptin, STAT phosphorylation was drastically

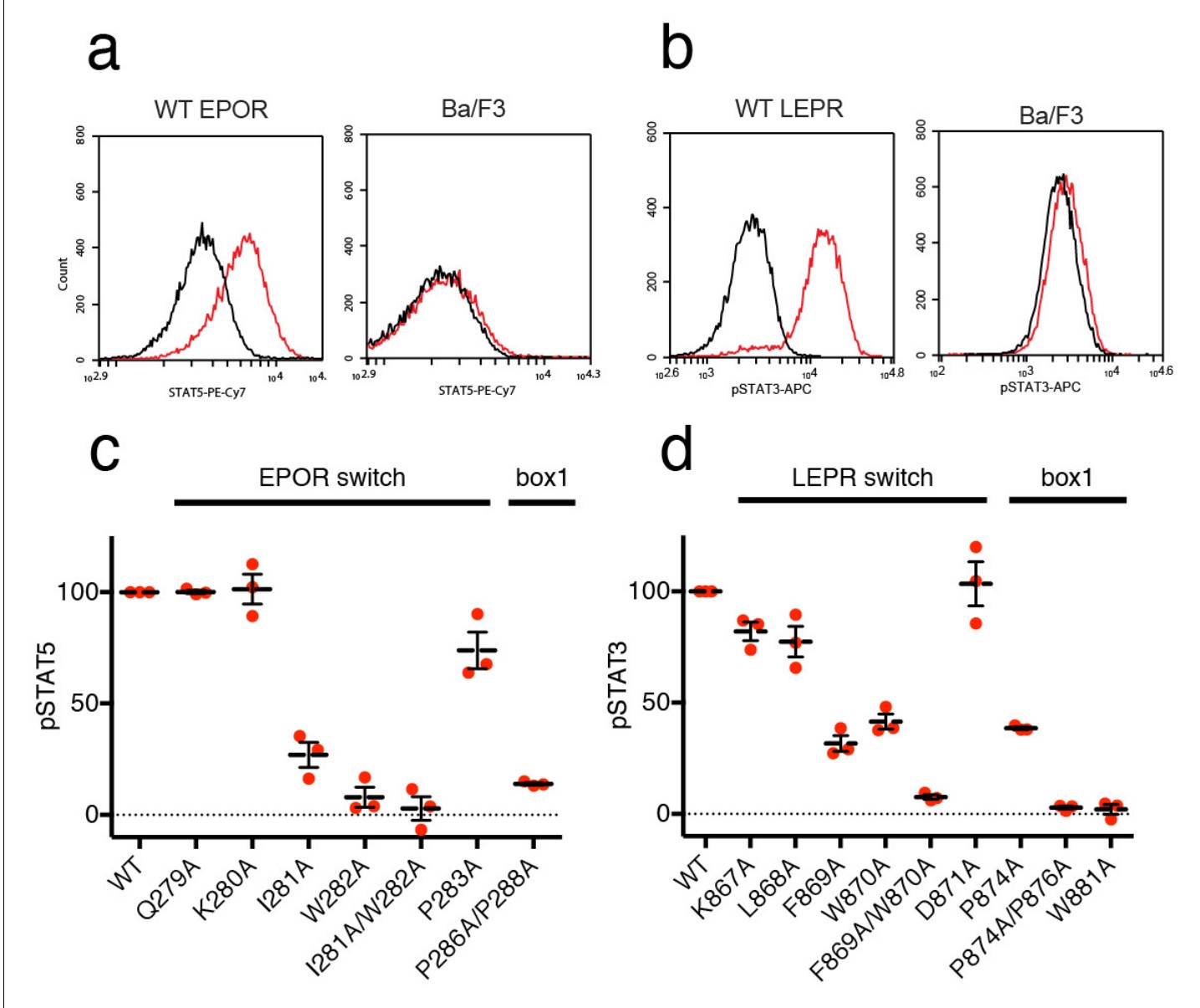

**Figure 5.** EPOR and LEPR switch residues are required for Epo and Leptin-induced STAT phosphorylation. (a,b) Stable Ba/F3 cell lines expressing wild-type, full length mouse EPOR or LEPR were generated and analyzed for STAT phosphorylation by flow cytometry after stimulation with EPO or leptin. (a) Representative plots comparing phospho-STAT5 staining of parental Ba/F3 cells or Ba/F3 cells expressing EPOR. Cells were stimulated with 1 nM mouse EPO for 15 min before fixation, staining, and analysis. (b) Representative plots comparing phospho-STAT3 staining of parental Ba/F3 cells or Ba/F3 cells expressing LEPR. Cells were stimulated with 100 pM mouse Leptin for 4 hr prior to fixation, staining, and analysis. (c) Analysis of STAT5 phosphorylation by flow cytometry for EPOR wild-type, switch region, and box1 mutants, stimulated with 1 nM EPO as in (a). Mean levels of STAT5 phosphorylation were assessed in three separate experiments, with wild-type signal representing 100% in all three experiments. (d) Analysis of STAT3 phosphorylation by flow cytometry for LEPR wild-type, switch region, and box1 mutants, stimulated with 100 pM Leptin, as in (b). Mean levels of STAT3 phosphorylation was assessed in three separate experiments, with wild-type signal representing 100% in all three experiments. Error bars represent standard error of the mean (SEM).

DOI: https://doi.org/10.7554/eLife.38089.011

The following figure supplement is available for figure 5:

**Figure supplement 1.** Analysis of Ba/F3 stable cell lines expressing EPOR or LEPR mutants.

DOI: https://doi.org/10.7554/eLife.38089.012

reduced (*Figure 5C–D* and *Figure 5—figure supplement 1*). We then tested cell lines containing alanine mutations across the membrane-proximal region N-terminal to box1, including the JAK2 switch residues in EPOR and LEPR (Ile281/Trp282 and Leu868/Phe869/Trp870, respectively) along with solvent-facing residues both N- and C-terminal to these contact sites. When switch residues Ile281 and Trp282 in EPOR were mutated, there was a significant drop in pSTAT5 phosphorylation, with an even stronger defect seen in the double Ile281/Trp282 to alanine mutant (*Figure 5C*). Non-contact residues Gln279 and Lys280 did not appreciably affect STAT5 phosphorylation, and mutation of Pro283 had only a minor effect. For LEPR, mutation of switch residues Phe869 and Trp870 reduced STAT3 phosphorylation more than 50%, while the double mutation of Phe869 and Trp870 to alanine reduced signaling to levels equivalent to those seen for box1 mutants (*Figure 5D*). Mutation of switch residue Leu868, which is also a contact residue, had a less substantial effect on STAT3 phosphorylation when compared to Phe869 and Trp870. Mutation of solvent-facing LEPR residues Lys867 or Asp871 had only a minor effect on STAT3 phosphorylation. Based on these experiments, we conclude that interactions between EPOR and LEPR switch region residues and the JAK2 FERM F3 subdomain on an opposing JAK2 molecule are required for STAT phosphorylation, indicating that our structures represent active JAK2-receptor dimer complexes.

## Discussion

Based on the studies presented here, we propose a model by which receptor-mediated intracellular dimerization of the JAK2 FERM–SH2 domains (*Figure 6A*) positions both JAK2 molecules in a conformation that enables activation of the kinase domains (*Figure 6B*). This model assumes that the C-terminus of the FERM–SH2 is facing the cytoplasmic side, so that the linked pseudokinase/kinase module would not sterically clash with the membrane (*Figure 6B*). The distances between the structurally resolved C-termini (residue Asn515) of the dimerized JAK2 FERM–SH2 domains in complex with EPOR and LEPR are 27 and 46 Å, respectively (*Figure 6—figure supplement 1*), indicating that the two SH2-linked pseudokinase domains would be close together upon dimerization. Given the short length of the linker between the JAK2 SH2 and pseudokinase (approximately 20 residues), this close apposition could produce a conformation capable of disrupting a pseudokinase/kinase auto-inhibitory complex (*Lupardus et al., 2014*; *Shan et al., 2014*) and permit trans-phosphorylation of the kinase activation loops to fully activate signaling. Currently, the transition of a JAK pseudokinase/kinase complex from an autoinhibited to active form is not well understood, but disruption of a trans-autoinhibited JAK2 pseudokinase/kinase dimer is also a possibility (*Brooks et al., 2014*; *Varghese et al., 2014*).

Our studies performed in stably transfected Ba/F3 cells show that disruption of the JAK2/EPOR or JAK2/LEPR dimer interfaces by mutation of the receptor switch residues renders the associated JAK2 molecules inactive, while mutation of these same residues (at least for JAK2/LEPR) does not significantly alter the in vitro receptor affinity for JAK2. The ability of these switch mutations to 'decouple' downstream STAT phosphorylation from JAK2/receptor binding is key evidence supporting the functional relevance of the dimeric JAK/receptor conformations seen in our structures. While STAT phosphorylation is generally regarded as a marker of JAK activation, it remains an indirect measure of JAK kinase activity. However, our data corroborate results published by Constantinescu and colleagues that describe the EPOR/JAK2 interaction (*Constantinescu et al., 2001*; *Huang et al., 2001*). In these studies, it was shown that mutation of EPOR switch region residues Ile257 and Trp258 was able to disconnect EPO-dependent JAK2 phosphorylation and cell growth from JAK2-dependent EPOR cell surface expression. Another study using IL5R-gp130 receptor chimeras to assess JAK1 activation demonstrated that mutation of a switch region tryptophan from the gp130 intracellular domain (Trp562) disrupted JAK1 signaling, yet did not affect JAK1 co-immunprecipitation with the chimeric receptor (*Haan et al., 2002*). These parallel results, obtained in three receptor systems utilizing two different JAKs, firmly substantiate the hypothesis that juxtamembrane switch residues play a key role in JAK activation.

The FERM domain is a remarkably prolific protein-protein interaction module (*Ferrao and Lupardus, 2017*), and the identification of new cytokine receptor interactions with the JAK2 PH-like (F3) subdomain adds to this compendium. The switch residue binding site on the F3 subdomain is typically recognized as the binding pocket for phosphatidyl-inositol triphosphate (PIP3) headgroups in PH domain-containing kinases such as AKT and BTK (*Baraldi et al., 1999*; *Thomas et al., 2002*). In

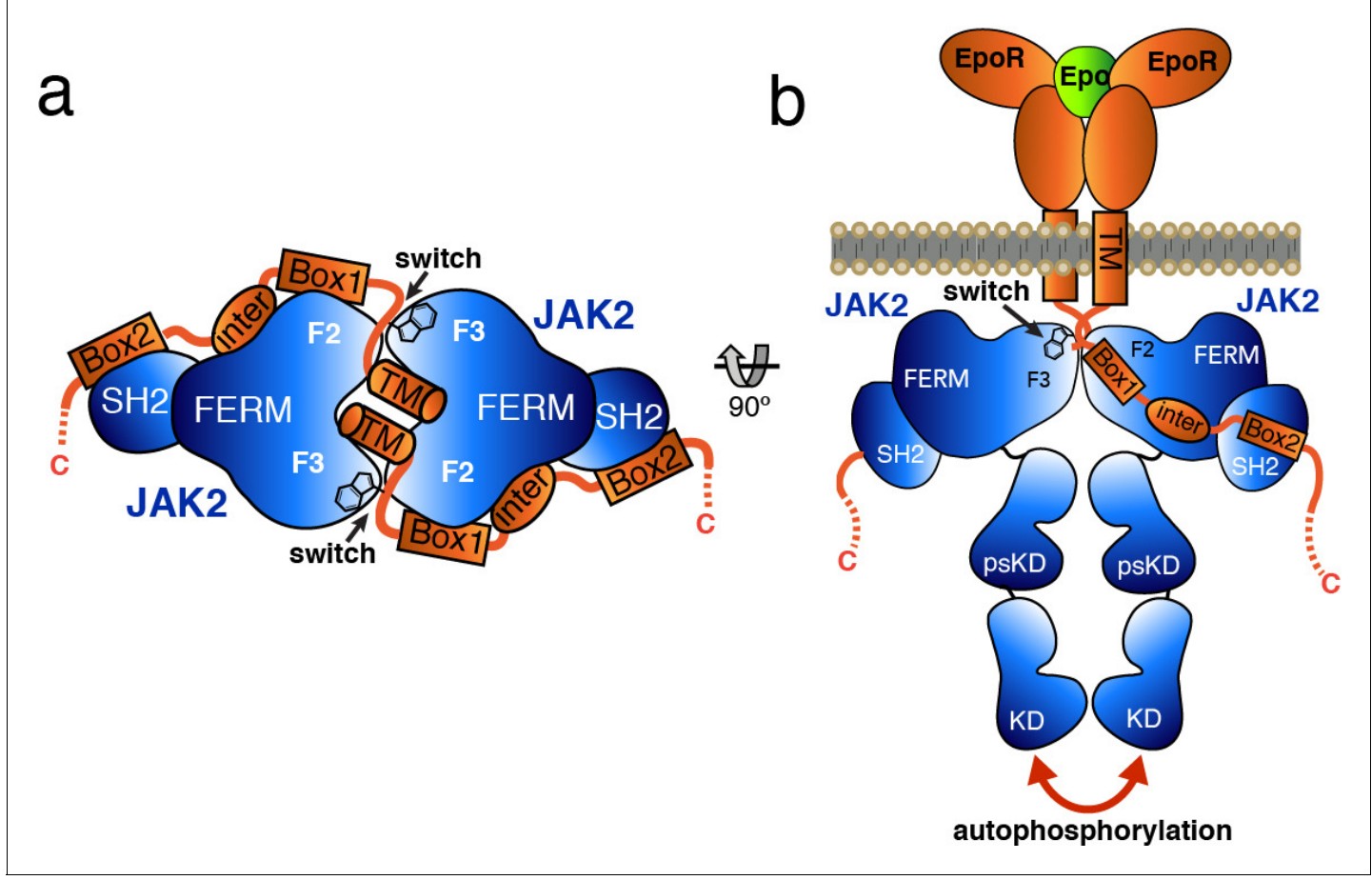

**Figure 6.** Model for JAK2 dimerization and activation. (**a**) Schematic diagram showing a top view of the JAK2/EPOR dimer, with the EPOR switch regions engaged and transmembrane (TM) domains displayed. (**b**) Schematic representation of the activation of JAK2 upon EPO-induced EPOR TM rearrangement and JAK2 dimerization.

DOI: https://doi.org/10.7554/eLife.38089.013

The following figure supplement is available for figure 6:

**Figure supplement 1.** Analysis of the JAK2/EPOR and JAK2/LEPR dimers.

DOI: https://doi.org/10.7554/eLife.38089.014

addition, Focal Adhesion Kinase, which contains a FERM domain, has also been shown to depend on its F3 subdomain to mediate dimerization and activation (*Brami-Cherrier et al., 2014*). FAK dimerization is mediated by symmetric exchange of tryptophan (Trp266) residues found in the F3 β1/β2 loop, with each tryptophan sidechain interacting with the opposite PH-like domain in a similar manner to EPOR and LEPR switch residues (*Figure 3—figure supplement 1*). Given the key role of the PH domains in function of a number of kinases, disruption of PH-mediated interactions by small molecule inhibitors has been proposed as a potential approach for inhibition (*Miao et al., 2010*). The mechanistic insights described here suggest targeting the switch-binding pocket on a JAK PH-like domain could be a novel approach for pharmacological inhibition of JAK signaling.

Cytokine-mediated reorientation of receptor transmembrane (TM) sequences into a specific 'activated' conformation has previously been suggested as an important trigger for JAK signaling (*Seubert et al., 2003*; *Brooks et al., 2014*; *Matthews et al., 2011*; *Defour et al., 2013*; *Brown et al., 2005*; *Staerk et al., 2011*). Recent studies of growth hormone receptor (GHR) suggest that growth hormone binding induces a transition from a parallel to a crossed-over TM dimer conformation which physically separates the JAK2-binding receptor box1 sequences (*Brooks et al., 2014*). Other work on EPOR and the thrombopoietin receptor (TPOR) has also suggested that TM dimer tilt angle may play a role in cytokine-induced JAK2 activation (*Defour et al., 2013*; *Seubert et al.,*

*2003*). In our structures, we find a Cα-Cα distance between the switch tryptophan residues in the EPOR (Trp283) and LEPR (Trp872) dimers of approximately 45 Å (*Figure 6—figure supplement 1*), suggesting a parallel TM dimer may not be sufficient to bridge the distance between these two interaction sites. Instead, a specific ligand-induced crossed-over or asymmetric TM dimer conformation could trigger switch region binding to the opposing F3 subdomain, and subsequent formation of the JAK2 dimer as we see in our structures. The requirement of a specific TM conformation for maximal JAK activation can also help explain data obtained using receptor dimerization methods such as antibodies (*Li et al., 2013*; *Kai et al., 2008*; *Müller-Newen et al., 2000*; *Zhang et al., 2012*), peptides (*Livnah et al., 1996*), diabodies (*Moraga et al., 2015*; *Nakano et al., 2009*), and engineered cytokines (*Moraga et al., 2017*; *Rafei et al., 2007*). While these non-natural means of dimerization do engage JAK signaling pathways, JAK kinase activity, as measured by receptor or STAT phosphorylation, is rarely induced at the same maximal levels seen for the native ligands. These findings correlate with studies that show that JAK activation is sensitive, but not entirely disrupted, by changing the juxtamembrane peptide length by adding alanine residues (*Constantinescu et al., 2001*; *Greiser et al., 2002*). Atypical TM conformations produced using surrogate dimerizing agents could plausibly alter the structure and kinetics of JAK dimer formation, producing unique levels of JAK activation and varied downstream signaling outputs (*Moraga et al., 2015*, *2017*; *Syed et al., 1998*).

A number of studies on the EPOR/JAK2 and gp130/JAK1 systems have suggested that the receptor juxtamembrane region takes on a specific helical conformation important for maximal JAK activation (*Constantinescu et al., 2001*; *Seubert et al., 2003*; *Li et al., 2014*; *Greiser et al., 2002*). While we do not see a helical conformation for the juxtamembrane peptides in our EPOR or LEPR structures, the receptor peptides used in our studies do not include a TM segment, which would likely anchor the juxtamembrane helices in place. The movement of a rigid pair of TM/juxtamembrane helices following cytokine engagement remains a plausible means for a ligand-bound receptor to enforce a specific conformation compatible with the JAK2 dimeric complexes presented here. While our structures cannot fully refute alternative scenarios such as a requirement for the switch residues in 'active state' dimerization of the receptor juxtamembrane regions, the evidence provided by our two dimeric JAK2 structures with similar architecture along with supporting functional data strongly suggests the presence of a JAK2 dimer in the context of an active cytokine receptor complex. A high-resolution structure of an active JAK-bound receptor signaling complex in the membrane will be required to fully describe the interplay between JAK and receptor subunits.

In our two structures, we see the JAK2/EPOR and JAK2/LEPR dimers interface at different angles. Currently, we are unable to ascertain whether these differing angles are due to crystal packing or are a result of differences in the receptor sequences between EPOR and LEPR. Regardless, the flexibility of the JAK2–JAK2 interface seen in these two examples suggests that contact between the two JAKs is not required for dimerization and that the receptor switch residues instead drive dimer formation. Given that most cytokine receptor signaling dimers consist not of homodimeric pairs, but instead involve two unique receptor chains uniting two unique JAKs (i.e. JAK1/JAK2), a lack of specific contacts between the dimerized JAKs may be a feature to preserve the modularity of the JAK system. Receptor-driven dimerization would allow for receptor sequence variation to fine-tune JAK dimer formation and binding affinity to regulate kinase signaling strength. Our working model presented here further illuminates the inherent modularity and flexibility found in the JAK-coupled receptor systems, which underlies their propagation and success as key signaling nodes in higher eukaryotes.

## Materials and methods

**Key resources table**

| Reagent type (species) or resource | Designation | Source or reference | Identification | Additional information |
|---|---|---|---|---|
| Biological sample (*Mus musculus*) | Erythropoietin | R and D systems | 959-ME | Concentration (1 nM) |

*Continued on next page*

*Continued*

| Reagent type (species) or resource | Designation | Source or reference | Identification | Additional information |
|---|---|---|---|---|
| Biological sample (*Mus musculus*) | Leptin | R and D systems | 498-OB | Concentration (100 pM) |
| Antibody (*Capra aegagrus*) | Anti-human leptin receptor polyclonal antibody | R and D systems | AF497 | Concentration (12.5 ng/uL) |
| Antibody (*Equus africanus*) | NorthernLights NL637-conjugated anti-goat monoclonal secondary antibody | R and D systems | NL002 | Dilution (1:200) |
| Biological sample (*Mus musculus*) | Epo-Fc fusion | Abcam | ab170076 | Concentration (12.5 ng/uL) |
| Antibody (*Mus musculus*) | Anti-human phospho-Stat3 monoclonal antibody | eBioscience/ ThermoFisher | 17-9033-41 | Dilution (1:40) |
| Antibody (*Mus musculus*) | Anti-human phospho-Stat5 monoclonal antibody | eBioscience/ ThermoFisher | 25-9010-42 | Dilution (1:40) |
| Other | Ni-NTA Superflow resin | Qiagen | 30430 | |
| Other | Glutathione Sepharose 4B resin | GE healthcare | 17075605 | |
| Other | Superdex 200 Hi-load 16/60 column | GE healthcare | 28989335 | |
| Chemical compound, drug | EDTA-free protease inhibitors | Roche | 11836170001 | |
| Commercial assay or kit | BirA biotinylation kit | Avidity | BirA500 | |
| Commercial assay or kit | QuikChange II XL Site-Directed Mutagenesis Kit | Agilent | 200522 | |
| Commercial assay or kit | ExiProgen ProXpress PCR Template Kit | Bioneer | K-7400 | |
| Commercial assay or kit | ExiProgen EC1 Protein Synthesis Kit | Bioneer | EK-77161 | |
| Chemical compound, drug | RPMI-1640 | produced in house | | |
| Chemical compound, drug | DMEM | produced in house | | |
| Chemical compound, drug | 1X Antibiotic-Antimycotic | Gibco | 15240062 | |
| Chemical compound, drug | 1% NEAA | Gibco | 11140050 | |
| Chemical compound, drug | TrypLE Express | Gibco | 12604013 | |
| Chemical compound, drug | FugeneHD | Promega | E2311 | |
| Chemical compound, drug | Retro-X Concentrator | Clontech/Takara | 631456 | |
| Chemical compound, drug | polybrene | Millipore | TR-1003-G | |
| Chemical compound, drug | Flow Cytometry Staining Buffer | ThermoFisher | 00-4222-26 | |

*Continued on next page*

*Continued*

| Reagent type (species) or resource | Designation | Source or reference | Identification | Additional information |
|---|---|---|---|---|
| Antibody (*Rattus norvegicus*) | anti-mouse CD16/CD32 monoclonal antibody (BD Fc Block) | BD Bioscience | 553141 | Dilution (1:200) |
| Chemical compound, drug | IC-fixation buffer | ThermoFisher | 00-8222-49 | |

## Protein expression and purification

To generate the single-chain JAK2/EPOR construct, an insert containing human EPOR box1/box2 (Ser273 to Cys338) linked to the C-terminus of human JAK2 (Asp36 to Thr514) with an 8xGly-Ser linker was cloned into a pAC-based vector in frame with an N-terminal His$_6$-TEV tag. Single-chain JAK2/EPOR baculovirus was then used to infect *T.ni* cells for 48 hr at 27°C. For the JAK2/LEPR complex, human JAK2 FERM–SH2 (Asp36 to Thr514) with an N-terminal His$_6$-TEV tag and human LEPR box1/box2 (Ser863 to Glu933) with a TEV cleavable N-terminal GST tag were cloned into pAC-based insect cell expression vectors. To obtain the JAK2/LEPR complex, *Sf9* cells were co-infected with JAK2 and LEPR baculoviruses and grown for 72 hr at 27°C.

To purify the JAK2/EPOR single chain, insect cells were harvested by centrifugation and resuspended in lysis buffer containing 50 mM Hepes pH 7.2, 500 mM NaCl, 10% glycerol, 1 mM TCEP and 20 mM imidazole supplemented with benzonase and EDTA-free protease inhibitor tablets (Roche). All subsequent steps were carried out at 4°C. Resuspended cells were homogenized, lysed by sonication, and subjected to centrifugation at 26,000 RCF. Lysate was incubated with Ni-NTA Agarose resin (Qiagen) in batch for 1 hr. Resin was recovered by centrifugation at 800 RCF, applied to a gravity column and washed with lysis buffer. Samples were eluted with lysis buffer supplemented with 300 mM imidazole. After IMAC elution, JAK2/EPOR was applied to a Superdex 200 Hi-load 16/600 equilibrated in SEC buffer (20 mM Hepes pH 7.2, 500 mM NaCl, 10% glycerol, 1 mM TCEP). Fractions containing JAK2-EPOR were pooled and the His$_6$ tag was cleaved overnight with TEV. The sample was then applied to Ni-NTA Agarose resin and washed with SEC buffer. Cleaved JAK2-EPOR eluted from the resin in SEC buffer supplemented with 40 mM imidazole. The sample was then concentrated and subjected to a final SEC run on a Superdex 200 Increase 10/300 equilibrated in either 10 mM Hepes pH 7.2, 200 mM NaCl, 1 mM TCEP, or 10 mM Citric acid pH 5.5, 200 mM NaCl, 1 mM TCEP. The protein was concentrated to 7 mg/mL for crystallography.

For purification of His-tagged JAK2 co-expressed with GST-LEPR, cells were harvested by centrifugation and resuspended in lysis buffer containing 50 mM Tris pH 8.5, 300 mM NaCl, 10% glycerol, 1 mM TCEP, 5 mM imidazole, and supplemented with PMSF, benzonase, and EDTA-free protease inhibitor tablets (Roche). Resuspended cells were homogenized, lysed by microfluidization, incubated with 0.2% CHAPS at 4°C for 1 hr, and subjected to centrifugation at 26,000 RCF. All subsequent steps were carried out at 4°C. Filtered lysate was incubated with Ni-NTA Superflow resin (Qiagen) in batch for 1 hr. Resin was recovered by centrifugation at 800 RCF, applied to a gravity column and washed with lysis buffer supplemented with 25 mM imidazole. Samples were eluted with lysis buffer supplemented with 300 mM imidazole. After IMAC elution, samples were concentrated and purified on a Superdex 200 Hi-load 16/600 column equilibrated in SEC buffer (25 mM Tris pH 8.5, 300 mM NaCl, 10% glycerol, 2 mM TCEP). Fractions containing JAK2/LEPR were pooled and tags cleaved overnight with TEV protease. Cleaved sample was applied to Ni-NTA Superflow resin and eluted with SEC buffer containing 40 mM imidazole. Eluted protein was applied to Glutathione Sepharose 4B resin (GE Healthcare) for removal of free GST. JAK2/LEPR was subjected to a final SEC run on a Superdex 200 Hi-load 16/600 column equilibrated in 25 mM Tris pH 8.5, 200 mM NaCl, 2% glycerol, 1 mM TCEP, followed by centrifugal concentration to 9 mg/ml for crystallography.

For Apo JAK2, the lysis and IMAC purification steps were performed as described for the JAK2/EPOR single chain construct. The IMAC elution was then applied to a Superdex 200 Hi-load 16/600 equilibrated in 20 mM Hepes pH 7.2, 500 mM NaCl, 1 mM TCEP. Fractions containing JAK2 were pooled and concentrated to 250 μM. Protein was supplemented with BSA, Tween-20, and Arginine pH 7.0 to final concentrations of 1 mg/mL, 0.05%, and 200 mM, respectively.

## Protein crystallization

Single-chain JAK2/EPOR purified into a final buffer of 10 mM Hepes pH 7.2, 200 mM NaCl and 1 mM TCEP was highly insoluble, with near complete precipitation observed upon concentration. Upon further inspection with phase contrast light microscopy, the precipitation was determined to be crystalline in nature. These microcrystals were pelleted by centrifugation and dissolved by addition of 100 mM Citric acid pH 5.5. Lowering the pH enabled concentration of the protein to 7 mg/mL. After sparse matrix screening, a single initial hit was obtained in 100 mM Tris pH 8.5, 8% PEG8000. Subsequent preparations of JAK2/EPOR were subjected to final SEC in 10 mM Na Citrate pH 5.5, 200 mM NaCl, 1 mM TCEP to reduce spontaneous crystallization during purification. Diffraction quality crystals were obtained by microseeding, using the Seedbead kit (Hampton Research) into 100 mM Tris pH 7.6, 2–4% PEG8000. Crystals were cryoprotected in mother liquor supplemented with 30% ethylene glycol.

For crystallization of the JAK2/LEPR complex, protein was concentrated to 9 mg/mL in final SEC buffer. Diffraction quality JAK2/LEPR crystals were obtained in 0.1 M MES pH 6.5, 0.2 M $MgCl_2$, 5–10% PEG4000, and 10% ethylene glycol using microseeding and PEG4000 dehydration up to 10% PEG4000. Crystals were cryoprotected in mother liquor with a final concentration of 25% ethylene glycol.

## Data collection and structure determination

Data for JAK2/EPOR was collected at ALS beamline 5.0.1, and data for JAK2/LEPR was collected at SSRL beamline 12–2. All data were collected under cryo-cooled conditions (100K) and processed with with XDS and XSCALE (*Kabsch, 2010*). Both structures were solved by molecular replacement with the program PHASER (*McCoy et al., 2007*) using JAK2 FERM–SH2 coordinates (*McNally et al., 2016*) as a search model (PDB: 4Z32). Both structures were refined by iterative rounds of simulated annealing, coordinate, and B-factor refinement using the Phenix package (*Adams et al., 2010*), followed by model building and adjustment using COOT (*Emsley and Cowtan, 2004*). The JAK2/EPOR model was refined at 2.65 Å to final R/R$_{free}$ statistics of 22.5/26.3%. Ramachandran statistics calculated by MolProbity (*Davis et al., 2007*) indicate that 96.4% of residues are in favored conformations. The final JAK2/EPOR model contains the following residues: JAK2 chain A (residues 35–277, 283–332, 336–515) and EPOR chain N (residues 279–335); JAK2 chain B (residues 37–277, 281–330, 337–515) and EPOR chain O (residues 279–317, 324–335); JAK2 chain C (residues 35–136, 140–277, 283–331, 337–441, 447–515) and EPOR chain M (residues 279–322); JAK2 chain D (residues 34–136, 140–277, 283–330, 337–415, 423–439, 449–465, 470–481, 498–514) and EPOR chain P (residues 276–322). The JAK2/LEPR model was refined at 2.83 Å to final R/R$_{free}$ statistics of 22.8/24.1%. Ramachandran statistics calculated by MolProbity (*Davis et al., 2007*) indicate that 95.6% of residues are in favored conformations. The final JAK2/LEPR model contains the following residues: JAK2 chain A (residues 38–47, 50–103, 110–276, 282–329, 338–515) and LEPR chain C (residues 866–885); JAK2 chain B (residues 41–47, 51–103, 110–240, 244–275, 284–329, 339–487, 490–514) and LEPR chain D (residues 868–885). Coordinates and structure factors have been deposited into the RCSB database as PDB ID 6E2Q (JAK2/EPOR) and PDB ID 6E2P (JAK2/LEPR). Structural figures were prepared with PyMOL (http://www.pymol.org/). Structural superpositions were carried out using the SSM algorithm from SUPERPOSE (*Maiti et al., 2004*).

## In vitro translation (IVT) of receptor peptides

Constructs encoding the cytoplasmic domain of various human cytokine receptors predicted to contain the Box1 and Box2 motifs were generated synthetically in the pIDT-SMART vector (IDT Technologies). Domain boundaries were as follows: EPOR (273-338), GHR (288-352), LEPR (863-933), IFNGR2 (272–337), GMCSFR (347-400), PRLR (259-319), TSLPR (253-318), IL5R (363-420), IL23R (377-447), IL3RA (326-378), GP130 (642-700), IFNLR1 (250–299). Mutant LEPR vectors were produced using the QuikChange II XL site-directed mutagenesis kit (Agilent). Templates for IVT were generated from these constructs by two rounds of PCR using the ExiProgen ProXpress PCR Template Kit. Each construct contained an N-terminal Avi tag and a C-terminal His$_6$ tag. IVT was carried out using the ExiProgen EC1 Protein Synthesis Kit supplemented with 1.5 μg BirA (Avidity) and 33.5 μL 500 μM Biotin.

## BioLayer interferometry

Products from IVT reactions were immobilized directly onto streptavidin biosensors. Apo JAK2 was purified according to the protocol described above. All JAK2 assays were performed in 20 mM Hepes pH 7.2, 500 mM NaCl, 1 mM TCEP, 1 mg/mL BSA, 0.05% Tween-20, 200 mM Arginine pH 7.0. Assays were performed in triplicate on an Octet Red384 (ForteBio) and double referenced against the buffer signal and a reference sensor. All LEPR peptide variants were loaded onto SA biosensors to a response of 0.5 nm. Binding of each LEPR variant to JAK2 was measured at JAK2 concentrations of 100.0, 33.3, 11.1, 3.70, 1.23, 0.41, and 0.14 μM. JAK2 association and dissociation steps were both 60 s, and in both cases rapid equilibrium was achieved. Double referenced response values at equilibrium were plotted as a function of concentration and fit to a global one site-specific binding model with a shared $R_{max}$ in Prism (Graphpad Software).

## Ba/F3 stable cell line generation

The mouse Ba/F3 cell line used in these studies was obtained from the Genentech in house cell repository (gCELL). Prior to batch release, the cells tested negative for mycoplasma and cross-species contamination. Ba/F3 cells were cultured in RPMI-1640 supplemented with 10% heat inactivated FBS, 2 mM Glutamine, 1X Antibiotic-Antimycotic (Gibco), 10 mM Hepes pH 7.2, and 10 ng/ml IL-3. Ba/F3 cells were grown to a maximum density of $2.0 \times 10^6$ cells/mL and split to a density $0.1 \times 10^6$ cells/mL. 293 T cells were grown in DMEM supplemented with 10% FBS, 1% NEAA (Gibco), 2 mM Glutamine, 10 mM Hepes pH 7.2, and 1X Antibiotic-Antimycotic. At 80% confluence, 293 T cells were dissociated using TrypLE Express (Gibco). Both cell lines were grown in the presence of 5% $CO_2$ at 37°C in cell culture flasks (Corning). Retroviral constructs encoding for WT murine Lepr (Met1-Val1162) or murine Epor (Met1-Ser507) with a C-terminal 3xFLAG tag were synthesized and cloned into the MigR1 vector. Mutations were obtained using the QuikChange II XL Site-Directed Mutagenesis Kit (Agilent). FuGENE HD (Promega) was used to transfect 25 μg of DNA at a 4:1 MigR1: pCL-eco ratio into $7 \times 10^6$ 293 T cells plated in a T150 flask 24 hr prior. Supernatant containing virus was harvested 72 hr post-transfection, precipitated using Retro-X Concentrator (Clontech) and resuspended in 2 mL Ba/F3 media. Concentrated viral supernatant was added to $1.4 \times 10^6$ Ba/F3 cells in a total volume of 4 mL and supplemented with 6 μg/ml polybrene (Millipore). Transduced cells were bulk sorted based on GFP expression with a FACSAria (BD Biosciences) and collected into Ba/F3 conditioned media filtered through a 0.22 μm membrane.

## Surface receptor staining

Surface expression of Lepr was assayed by staining cells using a mouse leptin receptor antibody (R and D Systems, AF497) followed by a NorthernLights NL637-conjugated secondary antibody (R and D Systems, NL002). Surface expression of Epor was assayed using an Epo-Fc fusion (Abcam) labeled with NHS-Cy5 (Sigma). $1 \times 10^6$ Ba/F3 cells expressing Epor or Lepr variants were washed 3x and resuspended in eBioscience Flow Cytometry Staining Buffer (ThermoFisher). Prior to all staining steps, the cells were blocked with anti-mouse BD Fc Block (BD Bioscience) for 30 min on ice. Cells were then incubated with murine Lepr antibody or Cy5-Epo-Fc fusion (2.5 μg/$10^6$ cells) for 30 min on ice. For Lepr expressing cells, this was followed by another 3x wash in flow cytometry staining buffer, reblocking with anti-mouse BD Fc Block, and a 30 min incubation with NL637-conjugated secondary antibody. All samples were subjected to a final 3x wash and resuspension in 300 uL flow cytometry staining buffer. Surface staining was analyzed using an Accuri C6 flow cytometer.

## Stat3/Stat5 phospho-flow cytometry

For the Lepr/Stat3 assay, $5 \times 10^6$ cells expressing wild-type or mutant Lepr receptor were washed 3x with PBS and incubated with 100 pM mouse Leptin (R and D Systems, 498-OB) in the absence of FBS and IL-3 for 4 hr at 37°C. For the Epor/Stat5 assay, $5 \times 10^6$ cells expressing wild-type or mutant Epor were starved of FBS and IL-3 for 4 hr at 37°C followed by stimulation with 1 nM mouse Erythropoietin (R and D Systems, 959-ME) for 15 min. After incubation with Leptin or Erythropoietin, cells were immediately fixed with 3 mL IC-fixation buffer (Thermo-Fisher) for 30 min at room temperature, spun down, and resuspended in 1 mL ice cold methanol. Cells were permeabilized by the addition of 1 mL ice cold methanol and 30 min incubation at 4°C. Fixed cells were stored overnight in methanol at −20°C. The next day, cells were pelleted, methanol removed, and cells resuspended and

washed 2X in Flow Cytometry Staining buffer (Thermo-Fisher). After the second wash, cells were resuspended in staining buffer, and blocked with 1 μL Mouse BD Fc Block (BD Bioscience) for 10 min at room temperature. Cells were then stained with 5 μL of phospho-Stat3 antibody (pStat3/APC, eBioscience) or phospho-Stat5 antibody (pStat5/PE-Cy7, eBioscience) for 60 min at room temperature in the dark. Cells were washed 3X and resuspended in flow cytometry staining buffer followed by analysis using an Accuri C6 flow cytometer (BD). To account for nonspecific staining, an unstimulated control was run alongside each experiment. The average fluorescence signal of the unstimulated cells was subtracted from the average signal of the cytokine-stimulated cells, followed by normalization to the wild-type variant. All assay steps were performed in triplicate.

## Acknowledgements

We thank the Biomolecular Engineering department at Genentech for assistance in cloning and protein expression, and Elizabeth Helgason for her assistance in cell-free peptide synthesis. This research used resources of the Advanced Light Source, which is a DOE Office of Science User Facility under contract no. DE-AC02-05CH11231. ALS Beamline 5.0.1 is funded in part by NIH grant S10OD021832. Use of the Stanford Synchrotron Radiation Lightsource, SLAC National Accelerator Laboratory, is supported by the US Department of Energy, Office of Science, Office of Basic Energy Sciences under Contract No. DE-AC02-76SF00515. The SSRL Structural Molecular Biology Program is supported by the DOE Office of Biological and Environmental Research, and by the National Institutes of Health, National Institute of General Medical Sciences (including P41GM103393).

## Additional information

### Competing interests

Ryan D Ferrao, Heidi JA Wallweber, Patrick J Lupardus: employee of Genentech, Inc. during the period when this work was performed.

### Funding

No external funding was received for this work.

### Author contributions

Ryan D Ferrao, Conceptualization, Data curation, Formal analysis, Validation, Investigation, Visualization, Methodology, Writing—original draft; Heidi JA Wallweber, Conceptualization, Investigation, Methodology; Patrick J Lupardus, Conceptualization, Data curation, Formal analysis, Supervision, Validation, Investigation, Methodology, Writing—original draft, Project administration, Writing—review and editing

### Author ORCIDs

Patrick J Lupardus (iD) http://orcid.org/0000-0002-8662-074X

### Decision letter and Author response

Decision letter https://doi.org/10.7554/eLife.38089.021
Author response https://doi.org/10.7554/eLife.38089.022

## Additional files

### Supplementary files

• Transparent reporting form
DOI: https://doi.org/10.7554/eLife.38089.015

### Data availability

Coordinates and structure factors have been deposited into the RCSB database as PDB ID 6E2Q (JAK2/EPOR) and PDB ID 6E2P (JAK2/LEPR).

The following datasets were generated:

| Author(s) | Year | Dataset title | Dataset URL | Database, license, and accessibility information |
|-----------|------|---------------|-------------|--------------------------------------------------|
| Ferrao RD, Wallweber HJA, Lupardus PJ | 2018 | Coordinates and structure factors for JAK2/EPOR | http://www.rcsb.org/structure/6E2Q | Publicly available at the RCSB Protein Data Bank (accession no. 6E2Q) |
| Ferrao RD, Wallweber HJA, Lupardus PJ | 2018 | Coordinates and structure factors for JAK2/LEPR | http://www.rcsb.org/structure/6E2P | Publicly available at the RCSB Protein Data Bank (accession no. 6E2P) |

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
