## [Decision Letter]

Thank you for submitting your article "Receptor-mediated dimerization of JAK2 FERM domains is required for JAK2 activation" for consideration by *eLife*. Your article has been reviewed by three peer reviewers, including Yibing Shan as the Reviewing Editor and Reviewer #1, and the evaluation has been overseen by John Kuriyan as the Senior Editor. The following individuals involved in review of your submission have agreed to reveal their identity: Jeffrey J Babon (Reviewer #2); Stefan Constantinescu (Reviewer #3).

The reviewers have discussed the reviews with one another and the Reviewing Editor has drafted this decision to help you prepare a revised submission.

Summary:

The authors report two crystal structures of the JAK2 FERM/SH2 domains, one with fragments of Leptin receptor (LEPR) bound and another with fragments of Erythropoietin receptor (EPOR) bound. Although homologous FERM/SH2 structures have been reported, these structures are the first of the prototypical JAK2 protein. The authors propose that the crystal dimers they captured, in which the FERM domains are cross-linked by the juxtamembrane region of the receptors, reflect the functional dimer structures in JAK2 signaling. Consistent with this notion, it was shown that mutations at LEPR juxtamembrane region abates JAK2 signaling but does not disrupt JAK2-LEPR binding. The reviewers think these results are potentially highly important, given that there is no clear structural understanding of JAK2 activation and this is the first plausible proposal with support from crystallographic data.

Essential revisions:

The reviewers raised several points that need to be addressed, which mostly concern the Discussion.

1) While this study offers elegant structural validation of previous functional studies which first proposed this switch motif in EPOR, as well as demonstrating that a single EPOR binds both JAK2 FERM domains within a JAK2 dimer, it is not clear whether this crystal structure necessarily represents a fully active structure, as claimed in the Discussion (fourth paragraph), or some intermediate dimer between fully active and inactive. Without including the kinase and pseudokinase domains, it is not obvious whether this arrangement of receptor and JAK is the same when the pseudokinase domain inhibits the kinase domain or not. It would be useful to explain in the last paragraph of the Introduction how the authors envisage intracellular changes in JAK2 conformation that are required to transition two JAK2 kinases to an active state.

2) The JAK2 SH2-JH2 linker is about four times as long as stated here (Discussion, first paragraph), maybe this increased inter-domain flexibility would change the model of KD/psKD auto-inhibitory complex described in this paragraph? Alternatively perhaps the authors referred to the short nJH2-JH1 linker.

3) This model implies a membrane orientation of the FERM/SH2 domains by which the prominently positively charged C-terminal helical subdomain of FERM will not be in an ideal position to contact the membrane; in the meantime, the extensive regions of the FERM/SH2 domains will be in membrane contact without much sign of positive electrostatic property. This raises question as to whether this dimer represents the stable active state of JAK2.

4) There could be other interpretations for the mutagenesis date at the receptor's juxtamembrane region. For instance, one alternative scenario could be that the receptor juxtamembrane regions are dimerized in the active state, and the mutations disrupt the dimerization. Excluding such possibilities would be beyond the scope of this manuscript, but they should be acknowledged and discussed.

---

## [Author Response]

Essential revisions:The reviewers raised several points that need to be addressed, which mostly concern the Discussion.1) While this study offers elegant structural validation of previous functional studies which first proposed this switch motif in EPOR, as well as demonstrating that a single EPOR binds both JAK2 FERM domains within a JAK2 dimer, it is not clear whether this crystal structure necessarily represents a fully active structure, as claimed in the Discussion (fourth paragraph), or some intermediate dimer between fully active and inactive. Without including the kinase and pseudokinase domains, it is not obvious whether this arrangement of receptor and JAK is the same when the pseudokinase domain inhibits the kinase domain or not. It would be useful to explain in the last paragraph of the Introduction how the authors envisage intracellular changes in JAK2 conformation that are required to transition two JAK2 kinases to an active state.

We appreciate the reviewers pointing this out, and have made adjustments to the Discussion (“active JAK2 dimer” changed to “JAK2 dimer”) to address the fact that these structures may not reflect the fully active state. We have also made a number of changes to the last paragraph of the Introduction to clarify our starting hypothesis at the outset of these studies.

2) The JAK2 SH2-JH2 linker is about four times as long as stated here (Discussion, first paragraph), maybe this increased inter-domain flexibility would change the model of KD/psKD auto-inhibitory complex described in this paragraph? Alternatively perhaps the authors referred to the short nJH2-JH1 linker.

We thank the reviewers for catching this error. Our analysis of the linker length (~10 amino acids as stated) was incorrect, and after careful review of our structures (JAK2 FERM/SH2) as well as analysis of PDB structures (4FVQ, Bandaranayake NSMB 2012) we have found that the ordered C-term of the JAK2 FERM/SH2 is Asn515 and ordered N-term of the JAK2 pseudokinase (psKD) is Val536, so the linker length is approximately 20 residues. We have adjusted our manuscript accordingly (Discussion, first paragraph). Interestingly, analysis of JAK1 FERM/SH2 and pseudokinase structures (PDB 5IXD and 4L00) shows the resolved C-term of the SH2 to be Glu563 and resolved N-term of pseudokinase to be Trp564, so there is a very short linker between the JAK1 SH2 and pseudokinase domains. For TYK2 FERM/SH2 and pseudokinase (PDB 4PO6 and 4WOV), the resolved C-term of the SH2 to be Gly566 and resolved N-term of pseudokinase to be Ser580 (~12 residue linker).

Based on these structures, the linkers found in JAK1, JAK2, and TYK2 vary between zero and 20 residues. In the absence of proper structural context in both the published and currently described individual domain structures, we postulate that these linker regions are likely not solvent-exposed, and instead probably form some sort of structural “glue” between the pseudokinase, kinase, and FERM/SH2 domains that aids in maintaining an autoinhibited complex. The length of the linker may also reflect differences in autoinhibition mechanism, i.e. intra- or inter-JAK autoinhibitory interactions between pseudokinase/kinase domains. Regardless, we expect that structural changes induced by the dimerization of JAK2 FERM/SH2 domains would disrupt this autoinhibited state, in particular if the autoinhibited state involves an interaction between the FERM/SH2 and the pseudokinase/kinase near this linker connection site.

3) This model implies a membrane orientation of the FERM/SH2 domains by which the prominently positively charged C-terminal helical subdomain of FERM will not be in an ideal position to contact the membrane; in the meantime, the extensive regions of the FERM/SH2 domains will be in membrane contact without much sign of positive electrostatic property. This raises question as to whether this dimer represents the stable active state of JAK2.

We thank the reviewers for their astute observation of the JAK surface electrostatics – during our analysis, we noticed as well that the positively charged patch on the FERM F2 subdomain would not be in primary contact with the membrane. However, it is our opinion that the autoinhibited off state is probably the primary state in which JAKs reside, and the positioning of this basic patch may be a reflection of this stable “resting” state for the JAK/receptor complex. One could also argue that if the activated, dimeric JAK molecular surface is optimally electrostatically coupled with the membrane this could increase complex stability, activation kinetics and therefore JAK signal strength, and ultimately have negative consequences for the cell. In summary, we feel that the absence of a highly basic interface doesn’t argue for or against this JAK2 dimer being a snapshot of the active complex.

4) There could be other interpretations for the mutagenesis date at the receptor's juxtamembrane region. For instance, one alternative scenario could be that the receptor juxtamembrane regions are dimerized in the active state, and the mutations disrupt the dimerization. Excluding such possibilities would be beyond the scope of this manuscript, but they should be acknowledged and discussed.

We thank the reviewer for suggesting this and while we think this is unlikely, we agree that we cannot rule it out. We have added a paragraph to the text to further discuss the possibility of TM/juxtamembrane helix formation and receptor dimer formation to the text to address this possibility (Discussion, fifth paragraph).